# Fatty acid-binding protein 4 is an independent factor in the pathogenesis of retinal vein occlusion

**Fumihito Hikage**[1]☯**, Masato Furuhashi**[2]☯**, Yosuke Ida**[1]*, **Hiroshi Ohguro**[1]**, Megumi Watanabe**[1]**, Soma Suzuki**[1]**, Kaku Itoh**[1]

**1** Departments of Ophthalmology, Sapporo Medical University School of Medicine, Sapporo, Japan,
**2** Departments of Cardiovascular, Renal and Metabolic Medicine, Sapporo Medical University School of Medicine, Sapporo, Japan

☯ These authors contributed equally to this work.
* funky.sonic@gmail.com

**Data Availability Statement:** All relevant data are within the manuscript and its Supporting Information files.

## Abstract

The main objective of current study was to identify the fatty acid-binding protein 4 (FABP4) expressed in both adipocytes and macrophages in vitreous fluid from patients with retinal vein occlusion (RVO). Patients with RVO (n = 14, CRVO; central RVO n = 5, BRVO; branch RVO n = 9) and non-RVO (macular hole or epiretinal membrane, n = 18) were surgically treated by a 25 or 27G vitrectomy. Undiluted vitreous fluid samples obtained as the result of surgery were subjected to enzyme-linked immunosorbent assays to measure the levels of FABP4 and vascular endothelial growth factor A (VEGFA).Data including ocular blood flow by laser speckle flow graphy (LSFG), height and weight, systemic blood pressures and several blood biochemistry values were collected. Among the LSFG mean blur rate (MBR) values of the optic nerve head (ONH) at baseline, MA (MBR of all area), MV (MBR of the vascular area), and MV-MT (MBR of the tissue area) were significantly decreased in patients with CRVO. The levels of V-FABP4 and V-VEGFA were relatively or significantly (P< 0.05) higher in the BRVO or CRVO patients compared to the non-RVO patients, respectively. A positive correlation (r = 0.36, P = 0.045) or a negative correlation (r = -0.51, P = 0.006) was observed between Log V-FABP4 and Log V-VEGF, or Log V-FABP4 and MV-MT at post-operative 1-week, respectively. Furthermore, neither of these factors were affected with respect to sex, body mass index and several clinical parameters that were collected, except that a positive correlation was observed for Log V-FABP4 with blood urea nitrogen. Stepwise multivariable regression analyses indicated that MV-MT at post-operative 1week was independently associated with Log V-FABP4 after adjustment for age and gender, and gender and Log V-FABP4 were independently associated with Log V-VEGFA after adjustment for age. The findings reported herein suggest that an independent factor, FABP4 may be synergistically involved in the pathogenesis of RVO with VEGFA.

**Funding:** The authors received no specific funding for this work.

**Competing interests:** The authors have declared that no competing interests exist.

## Introduction

Retinal vein occlusion (RVO), a common retinal vascular disease with a prevalence of 0.52%–0.77% in adults, is caused by a thrombosis in the central (CRVO), hemi-central (hemi-CRVO), or branch retinal veins (BRVO) [1, 2]. In CRVO or hemi-CRVO, such an obstruction usually occurs at the level of the lamina cribrosa, while in BRVO, it typically occurs at an intersection of a branched central retinal vein with branched central retinal artery [3–5]. A major cause of RVO-associated visual impairment is due to macular edema, retinal ischemia, and neovascular complications [6]. Such clinical manifestations of RVO are largely related to elevated levels of vascular endothelial growth factor (VEGF) in the vitreous and retina due to retinal ischemia [3]. Based upon this evidence, several randomized clinical trials, such as BRIGHTER, and VIBRANT, demonstrated that the intravitreal injections of anti-VEGF agents substantially improved ME and visual acuity (VA) [7–10]. Despite this anti-VEGF therapy, RVO patients often complain about a decreased quality of vision due to symptomatic metamorphopsia, even though VA and ME are improved [11–15]. Therefore, to avoid this issue, other promising therapeutics other than VEGF should be investigated. In fact, several epidemiological studies have also revealed that hypertension (HT) is the strongest risk factor for RVO [1, 16–20] as well as other systemic diseases, including cardiovascular disease, diabetes mellitus (DM), hyperlipidemia, hyper-homocysteinemia, blood coagulation disorders, systemic inflammatory disorders, glaucoma, short axial length, and high body mass index [4, 5, 21–23]. Nevertheless, with the exception of VEGF, no significant demographic systemic factors including age or male gender, systemic factors including vascular risk factors or high levels of blood hematocrit, as well as several ocular factors including macular pigmentary change, epiretinal-membrane formation following long-standing macular edema (ME), retinociliary collaterals, and glaucoma have been reported to be associated with this risk [24].

Fatty acid-binding proteins (FABPs), intracellular lipid chaperones, are a group of molecules that coordinate lipid responses in cells [25–27]. Functionally, FABPs have the ability to reversibly bind hydrophobic ligands such as long-chain, saturated and unsaturated fatty acids with a high affinity [25–27]. It has been reported that FABPs stimulate the transport of lipids to specific cellular compartments, such as the endoplasmic reticulum indicating that they are involved in signaling, trafficking, and membrane synthesis, to mitochondria or peroxisomes for oxidation, to cytosolic or other enzymes to regulate their activity, to the nucleus for lipid-mediated transcriptional regulation, and to lipid droplets for storage [25–27]. Among the FABP family members, FABP4, alternatively referred to as adipocyte FABP (A-FABP) or aP2, which is present in both adipocytes and macrophages, can be detected in most body fluids and reflects several pathogenic states. In fact, elevated serum concentrations of FABP4 are known to be associated with obesity [28], insulin resistance [29], hypertension (HT) [30], dyslipidemia [31], atherosclerosis [32], renal dysfunction [33], purine metabolism [34], heart failure and cardiovascular events [35]. In addition, recent observations have also demonstrated that the concentration of FABP4 could be altered by administering therapeutic drugs for HT, dyslipidemia and DM [29–31]. Since these systemic diseases are known to be risk factors for RVO as above, these collective observations rationally suggest that FABP4 may also be involved in the pathogenesis of RVO. However, as of this writing, in terms of ocular FABPs, FABB5, also known as epidermal FABP, has only been detected within the lens [36].

In the current study, to elucidate the pathological involvement of FABP4 within the RVO, we surgically collected vitreous specimens from patients with RVO or non-RVO (epiretinal membranes or macular holes) and measured the FABP4 and VEGF concentrations in these samples.

## Methods

This study conformed to the principles outlined in the Declaration of Helsinki and was performed with the approval of the institutional ethical committee of Sapporo Medical University School of Medicine (282–76). Written informed consent was received from all of the participating subjects.

### Patients

Thirty two patients who had been consecutively operated on (n = 14 eyes) for RVO (mean age 67 ± 15 years; 6 male and 8 female, BRVO; 9 eyes, mean age 69 ± 13 years; 4 male and 5 female, CRVO 5 eyes, mean age 65 ± 19 years; 2 male and 3 female) and 18 non-RVO patients (mean age 68 ± 8 years; 6 male and 12 female) with a macular hole (n = 6 eyes) or an epiretinal membrane (n = 12 eyes) requiring a vitrectomy were recruited from the Muroran municipal hospital during Jan to Dec, 2017. In order to determine a suitable surgical indication of vitrectomy, all patients underwent a complete ophthalmologic evaluation before surgery with a best-corrected visual acuity (BCVA) determination, slit-lamp examination, fundus examination, intraocular pressure measurement, gonioscopy, and optical coherence tomography. A clinical preoperative and intraoperative assessment of disease activity was performed by one experienced retina specialist (K.I). The RVO diagnosis was based on flame-shaped retinal hemorrhages distributed in occluded retinal veins, with conventional multimodal imaging: color fundus photography, fluorescein angiography, and SD-OCT (Topcon DRI OCT Triton). The exclusion criteria were high myopia (> 6 diopters), and preoperative treatment for ME via injections of intravitreal anti-VEGF. In all patients, 25 or 27-gauge three-port pars plana vitrectomies were performed (Alcon Constellation Vision System), along with simultaneous cataract surgery under systemic anesthesia. Inter limiting membrane pealing, or air or 10–20% SF6 gas tamponade was performed for 3 eyes for RVO and 14 for non-RVO eyes during the surgery, respectively. Of the 14 eyes from RVO patients, 12 were associated with vitreous hemorrhage and others was with traction retinal detachment prior to the surgery. No serious postoperative complications were except for slight vitreous hemorrhaging and none of the eyes have required reoperations as of this writing. Data regarding each patient's general conditions were obtained from the patient and from the patient's general practitioner.

Medical check-ups, including body height and weight measurements, and the collection of peripheral blood specimens were performed after an overnight fast. After measuring anthropometric parameters, blood pressure was measured with subjects in a seated resting position, and the average blood pressure was used for the analysis. Peripheral venous blood samples were collected and a complete blood count and biochemical analyses were carried out.

### Biochemical measurements

Undiluted vitreous samples were obtained during the initial core vitrectomy from 14 RVO and 18 non-RVO subjects, who underwent vitrectomy. During the collection of the vitreous specimens, extreme care was exercised in terms of avoiding contamination of the samples with extraocular blood. These specimens were then immediately stored at -80 $^{o}$C until used in the analyses. The concentrations of vitreous FABP4 (V-FABP4) or VEGFA (V-VEGFA) were measured using commercially available enzyme-linked immunosorbent assay kits for FABP4 (Biovendor R&D, Modrice, Czech Republic) or human VEGFA (Fuji film Wako. Co., Japan). The intra- and inter-assay coefficients of variation in the kits were <5%. Protein concentrations of the vitreous specimens were determined using a commercially available kit (Pierce BCA Protein Assay Kit, Pierce Biotechnology, Rockford USA) according to the manufacturers

protocol. Levels of V-FABP4 and V-VEGFA were adjusted by vitreous protein concentration and were expressed as ng/mg protein and pg/mg protein, respectively.

Plasma glucose levels were determined by the glucose oxidase method. Hemoglobin A1c (HbA1c) was determined by a latex coagulation method and is expressed by the National Gly-cohemoglobin Standardization Program (NGSP) scale. Creatinine (Cr), blood urea nitrogen (BUN), uric acid, aspartate transaminase (AST), alanine aminotransferase (ALT), γ-glutamyl transpeptidase (γ-GTP) and lipid profiles, including total cholesterol and triglycerides, were determined by enzymatic methods. High-sensitivity C-reactive protein (hsCRP) was measured by a nephelometry method. As an index of renal function, the estimated glomerular filtration rate (eGFR) was calculated by an equation for Japanese subjects: eGFR (ml/min/1.73 m2) = 194 × creatinine (−1.094) × age (−0.287) × 0.739 (if female).

## Laser speckle flowgraphy (LSFG)

The images of the speckle contrast pattern (LSFG) produced by interference as a laser beam was scattered by erythrocytes moving through the ocular fundus vessels were obtained by a fundus camera equipped with an 830 nm diode laser and a charge-coupled device sensor (750 × 360 pixels) (LSFG-NAVI; Softcare Co, Ltd., Fukuoka, Japan) as described previously [37–39]. The acquired LSFG images were continuously monitored at 30 frames/sec over a 4-s period and averaged to produce a composite map of ocular blood flow. As an indicator of ocular blood flow at a specific site, the mean blur rate (MBR), in arbitrary units (AU), was calculated and those at several sites were reconstituted to form a 2-dimensional color-coded map of blood flow velocity. In the current study, we investigated the MBR of the optic nerve head (ONH) of following four categories; 1) MA; all area, 2) MV; the vascular area including the effects of choroidal vessels, 3) MT; the tissue area, and 4) MV-MT (to subtract the effects of choroidal vessels from MV). All measurements were performed in triplicate and the mean MBR value was calculated. Eye positions were continuously monitored during LSFG analysis with an auto tracking function, confirm that the same area was captured again during subsequent examinations.

## Statistical analysis

Numeric variables are expressed as the mean ± SD for normal distributions or medians (interquartile ranges) for skewed variables. Intergroup differences in percentages of demographic parameters were examined by the chi-square test. Comparison between two groups was done with the Mann-Whitney's U test. The distribution of each parameter was tested for its normality using the Shapiro-Wilk W test, and non-normally distributed parameters were logarithmically transformed for regression analyses. Correlations between two continuous variables were analyzed using Pearson's correlation coefficient. A p value of < 0.05 was considered to be statistically significant. Stepwise and subsequent multivariable regression analyses were performed to identify independent determinants of plasma XOR activity using age, gender and the variables with a significant after consideration of multicollinearity, and with the t-ratio calculated as the ratio of the unstandardized regression coefficient and the SE of the unstandardized regression coefficient, the standardized regression coefficient (β), the percentage of variance in the object variables that the selected independent predictors explained ($R^2$), and the Akaike information criterion (AIC). Of the candidate models, the best-fit model using AIC for each dependent variable was selected. P<0.05 was considered to be statistically significant. All data were analyzed using JMP 14.3.0 for Macintosh (SAS Institute, Cary, NC).

## Results

Table 1 and S1 Table provide information concerning the characteristics of the backgrounds of patients with RVO (n = 20) (RVO; n = 9, CRVO; n = 5) and non-RVO (n = 18) (macular hole; n = 7, epiretinal membrane; n = 11) including sex, age, body mass index, systemic and diastolic blood pressure, blood chemistry values including total cholesterol, triglycerides, fasting glucose, Hb A1c, BUN, Cr, eGFR, uric acid, AST and ALT, γGTP and hsCRP, and four LSFG ocular blood flow indexes including MA; mean blur rate (MBR) of the all of the ONH, MV; MBR of the vascular area of the ONH, MT; MBR of the tissue area of the ONH, and MV-MT. Among two RVO patient groups and the non-RVO (Table 1), MA was significantly decreased in the RVO patients compared to the non-RVO patients, and among three BRVO, CRVO and non-RVO groups (S1 Table), MA, MV and MV-MT were marked decreased in the case of the CRVO patients, compared to the others. Except that no significant difference was observed among the groups of patients.

In terms of the levels of vitreous FABP4 (V-FABP4) or VEGFA (V-VEGFA), both were significantly elevated in patients with RVO compared to those with non-RVO (P<0.05) (Fig 1A and 1B). In the comparison of the three groups; non-RVO, BRVO and CRVO, both V-FABP4

**Table 1. Characteristics of the patients with non-RVO and RVO (n = 32).**

|  |  | All | non-RVO | RVO | P |
|---|---|---|---|---|---|
| n |  | 32 | 18 | 14 |  |
| Sex (Male/Female) |  | 12/20 | 6/12 | 6/8 | 0.581 |
| Age (years) |  | 68 ± 11 | 68 ± 8 | 67 ± 15 | 0.951 |
| Body mass index |  | 23.8 ± 3.0 | 23.2 ± 3.4 | 24.6 ± 2.3 | 0.194 |
| Systolic blood presure (mmHg) |  | 137 ± 16 | 137 ± 17 | 138 ± 16 | 0.816 |
| Diastolic blood pressure (mmHg) |  | 80 ± 10 | 80 ± 10 | 80 ± 11 | 0.974 |
| Biochemical data |  |  |  |  |  |
|  | Total choleterol (mg/dL) | 204 ± 33 | 208 ± 40 | 198 ± 20 | 0.388 |
|  | Triglycerides (mg/dL) | 155 (103–220) | 120 (96–222) | 188 (120–214) | 0.323 |
|  | Fasting glucose (mg/dL) | 111 (99–134) | 115 (101–147) | 107 (90–122) | 0.143 |
|  | Hemoglobin A1c (%) | 6.0 ± 0.7 | 6.1 ± 0.9 | 6.0 ± 0.4 | 0.666 |
|  | Blood urea nitrogen (mg/dL) | 15 ± 5 | 15 ± 4 | 14 ± 5 | 0.536 |
|  | Creatinine (mg/dL) | 0.7 (0.6–0.8) | 0.7 (0.6–0.8) | 0.7 (0.6–0.8) | 0.874 |
|  | eGFR (mL/min/1.73m$^2$) | 71.7 ± 1.2 | 71.0 ± 17.4 | 72.5 ± 14.6 | 0.800 |
|  | Uric acid (mg/dL) | 5.0 ± 1.2 | 5.3 ± 1.2 | 4.7 ± 1.3 | 0.191 |
|  | AST (IU/L) | 24 (19–31) | 26 (20–33) | 22 (18–31) | 0.158 |
|  | ALT (IU/L) | 23 (15–28) | 24 (16–29) | 21 (14–29) | 0.518 |
|  | γGTP (IU/L) | 29 (16–53) | 26 (15–61) | 36 (19–54) | 0.392 |
|  | hsCRP (mg/dL) | 0.09 (0.04–0.13) | 0.06 (0.03–0.12) | 0.10 (0.05–0.14) | 0.380 |
| Laser speckle flowgraphy |  | [n = 27] | [n = 18] | [n = 9] |  |
|  | MA | 19.3 ± 6.0 | 21.0 ± 5.8 | **15.7 ± 4.7** | **0.025** |
|  | MV | 33.9 ± 9.4 | 36.0 ± 7.4 | 29.7 ± 11.9 | 0.102 |
|  | MT | 12.0 ± 3.3. | 12.7 ± 3.3 | 10.6 ± 3.1 | 0.124 |
|  | MV-MT | 21.9 ± 7.2 | 23.3 ± 5.6 | 19.1 ± 9.5 | 1.159 |
|  | MM | 8.5 ± 4.4 | 9.1 ± 4.8 | 7.1 ± 3..3 | 0.269 |

Variables are expressed as number, means ± SD or medians (interquartile ranges).

AST, aspartate transaminase; ALT, alanine transaminase; eGFR, estimated glomerular filtration rate; γGTP, γ-glutamyl transpeptidase; hsCRP, high-sensitivity C-reactive protein; MA, mean blur rate (MBR) of all area of optic nerve head (ONH); MV, MBR of the vascular area of ONH; MT, MBR of the tissue area of ONH.

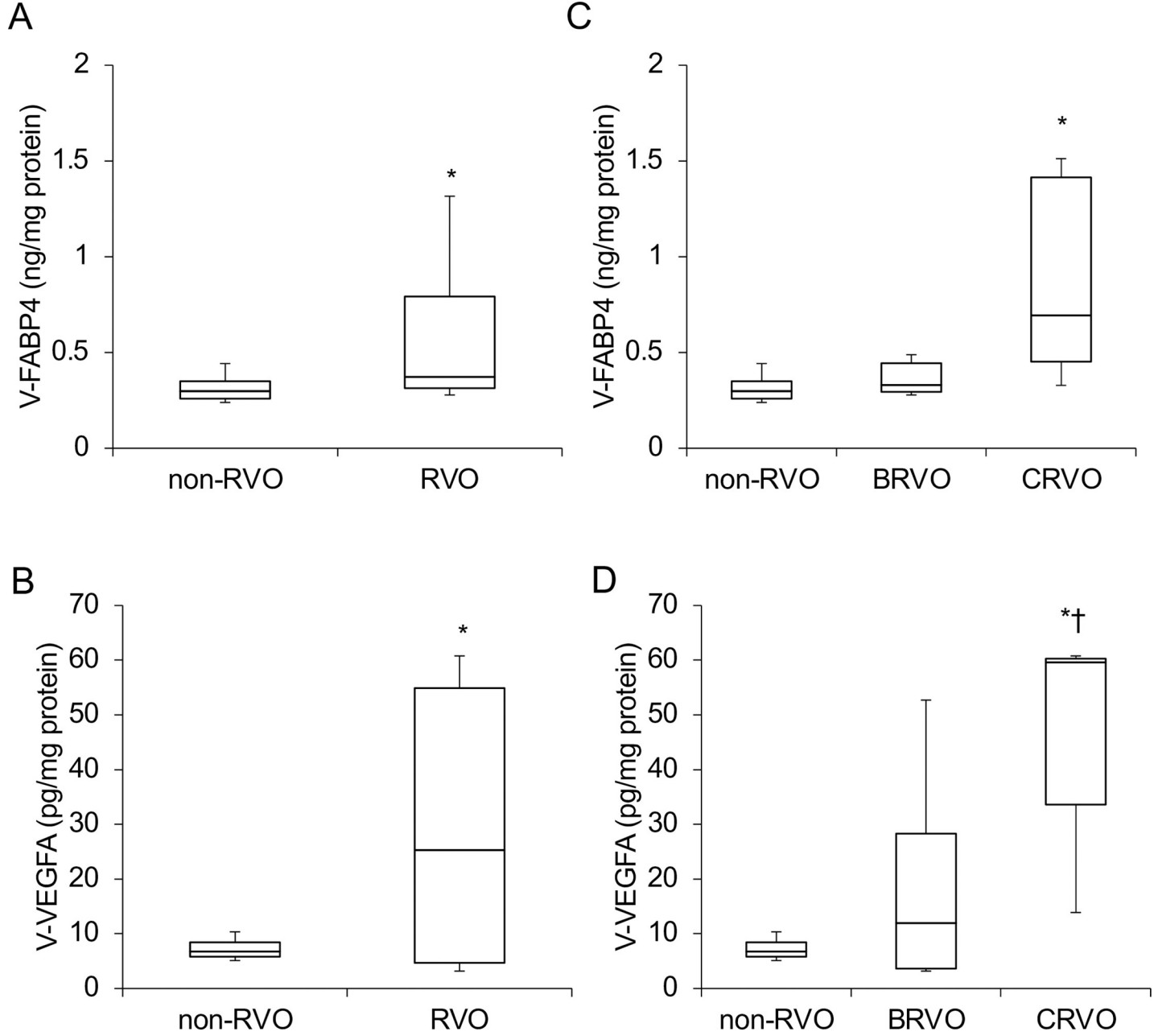

**Fig 1. Concentrations of vitreous FABP4 (V-FABP4) and VEGFA (V-VEGFA) in patients with non-RVO and RVO.** Undiluted vitreous specimens obtained surgically from patients with non-RVO (n = 18) and RVO (n = 14, BRVO; n = 9, CRVO; n = 5) were subjected to Enzyme-Linked Immuno-Sorbent Assay (ELISA) for FABP4 and VEGFA. The levels of V-FABP4 (ng/mg protein) and V-VEGFA (pg/mg protein) in the two groups; RVO and non-RVO groups (panels A and B), and three groups; BRVO, CRVO and non-RVO groups (panels C and D) were plotted. FABP4, fatty acid-binding protein 4; VEGFA, vascular endothelial growth factor A; V-FABP4, vitreous FABP4; V-VEGFA, vitreous VEGFA, RVO; retinal vein occlusion, BRVO; branch retinal vein occlusion, CRVO; central retinal vein occlusion. *P < 0.05 vs. non-RVO, †P < 0.05 vs. BRVO.

and V-VEGFA in the BRVO and CRVO groups were relatively or significantly (P<0.05) elevated in comparison with non-RVO group (Fig 1C and 1D). In addition, a positive correlation between Log V-FABP4 and Log V-VEGFA was observed (Fig 2 panel A, r = 0.36 P = 0.045). Since it is well known that FABP4 and VEGFA are closely associated with local blood circulation, post-operative ocular blood flow by LSFG was determined in order to elucidate the

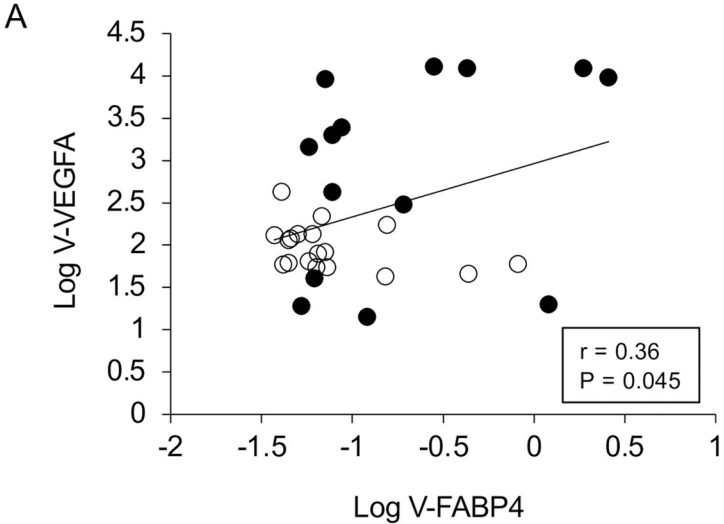

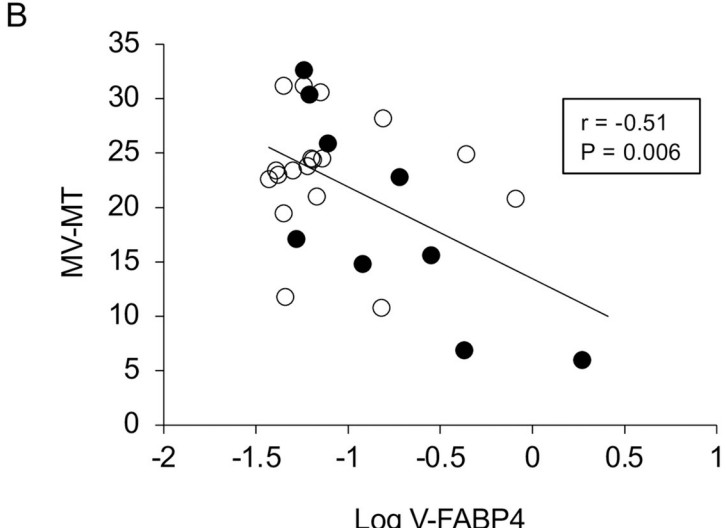

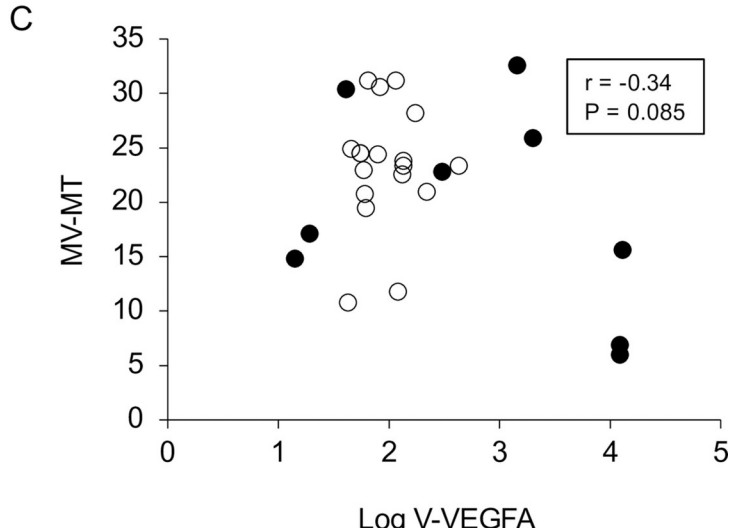

**Fig 2.** Correlations between Log V-FABP4 and Log V-VEGFA (A), and ocular blood flow (MV-MT at post-operative 1-week) and Log V-FABP4 (B) or Log V-VEGFA (C). Levels of Log V-FABP4 were plotted against Log V-VEGFA for each subject panel A, n = 32, r = 0.36, P = 0.045). Those of MV-MT at the post-operative 1-week for each subject (n = 32) were plotted against Log V-FABP4 (panel B, r = -0.51, P = 0.006) or Log V-VEGFA (panel C, r = -0.34, P = 0.085). Open circles, subjects with non-RVO; closed circles, subjects with RVO; FABP4, fatty acid-binding protein 4; VEGFA, vascular endothelial growth factor A; V-FABP4, vitreous FABP4; V-VEGFA, vitreous VEGFA, mean blur rate; MBR, optic nerve head; ONH, MV; the MBR of the vascular area of the ONH, MT; the MBR of the tissue area of the ONH.

clinical and pathological significance of V-FABP4 and V-VEGFA in RVO. As shown in Fig 2 panels B and C, and Table 2, Log V-FABP4 was negatively correlated with MV (r = -0.46 P = 0.015) and MV-MT (r = -0.51 P = 0.006) at post-operative 1-week. While in contrast, no significant correlation of Log V-VEGFA with the measured LSFG indexes was observed. These data indicated that V-FABP4 is significantly correlated with post-operative ocular blood circulation, and therefore this may be useful for evaluating a postoperative visual prognosis.

To study this conclusion further, correlation analyses between the vitreous concentrations of both factors and several clinical parameters indicated that Log V-FABP4 was positively correlated with Log V-VEGFA (r = 0.36, P = 0.045) and BUN (r = 0.37, P = 0.036) while, in contrast, Log VEGFA was positively correlated only with Log V-FABP4 (r = 0.36, P = 0.045) (Table 3). In addition, stepwise and subsequent multivariable regression analyses for Log V-FABP4 using age, gender, Log V-VEGFA and MV-MT as possible determinants indicated that MV-MT was independently associated with Log V-FABP4 after adjustment for age, gender and Log V-VEGFA. In contrast, similar analysis for Log V-VEGFA using age, gender and Log V-FABP4 as possible determinants indicated that gender and Log V-FABP4 were independently associated with Log V-VEGFA after adjustment for age (Table 4). These collective findings suggest that an independent factor, FABP4 may function to regulate ocular blood circulation, and be synergistically involved in the molecular pathology of RVO with VEGFA.

## Discussion

Upon retinal hypoxia, VEGF is expressed in various cells within the retina including retinal glial cells, retinal pigment epithelial cells, and vascular endothelial cells [40], and in turn, VEGF increases vascular permeability and promotes the proliferation of endothelial cells [41, 42]. In terms of the intraocular VEGF level, it was reported that those levels are not necessarily elevated in all RVO patients [43], and are widely varied among patients. In fact, it was reported that the intraocular VEGF level was significantly correlated with the retinal non-perfused area [44], as well as the severity of macular edema [43]. Based upon these findings, it would appear

**Table 2. Correlation analyses for Log V-FABP4 and Log V-VEGFA with blood flow at post-operative 1-week (n = 27).**

|  | Log V-FABP4 | | Log V-VEGFA | |
|---|---|---|---|---|
|  | **r** | **P** | **r** | **P** |
| MA | -0.35 | 0.071 | -0.24 | 0.232 |
| MV | **-0.46** | **0.015** | -0.27 | 0.176 |
| MT | -0.20 | 0.322 | -0.03 | 0.895 |
| MV-MT | **-0.51** | **0.006** | -0.34 | 0.085 |

MA, mean blur rate (MBR) of all area of optic verve head (ONH); MV, MBR of the vascular area of ONH; MT, MBR of the tissue area of ONH; V-FABP4, fatty acid-binding protein 4 in vitreous humor; V-VEGFA, vascular endothelial growth factor A in vitreous humor.

**Table 3. Correlation analyses for Log V-FABP4 and Log V-VEGFA (n = 32).**

| | Log V-FABP4 | | Log V-VEGFA | |
|---|---|---|---|---|
| | **r** | **P** | **r** | **P** |
| Age | 0.40 | 0.402 | -0.10 | 0.578 |
| Log V-FABP4 | - | - | **0.36** | **0.045** |
| Log V-VEGF | **0.36** | **0.045** | - | - |
| Body mass index | 0.11 | 0.539 | -0.07 | 0.705 |
| Systolic blood pressure | 0.11 | 0.542 | -0.08 | 0.669 |
| Diastolic blood pressure | -0.29 | 0.111 | -0.07 | 0.686 |
| Log AST | -0.22 | 0.229 | -0.22 | 0.229 |
| Log ALT | -0.11 | 0.560 | 0.04 | 0.843 |
| Log γGTP | -0.02 | 0.933 | 0.07 | 0.699 |
| BUN | **0.37** | **0.036** | -0.01 | 0.969 |
| Log Creatinine | 0.03 | 0.891 | 0.16 | 0.385 |
| eGFR | -0.10 | 0.603 | 0.08 | 0.683 |
| Uric acid | -0.18 | 0.329 | 0.08 | 0.675 |
| Total cholesterol | -0.12 | 0.522 | -0.16 | 0.392 |
| Log Triglycerides | 0.12 | 0.519 | 0.12 | 0.528 |
| Log Fasting glucose | 0.04 | 0.832 | -0.29 | 0.102 |
| Hemoglobin A1c | 0.02 | 0.926 | -0.10 | 0.573 |
| Log hsCRP | 0.12 | 0.509 | 0.11 | 0.555 |

AST, aspartate transaminase; ALT, alanine transaminase; eGFR, estimated glomerular filtration rate; γGTP, γ-glutamyl transpeptidase; hsCRP, high sensitivity C-reactive protein; V-FABP4, fatty acid-binding protein 4 in vitreous humor; V-VEGFA, vascular endothelial growth factor A in vitreous humor.

that the expression of VEGF initially increases after RVO due to retinal hypoxia caused by a vascular occlusion, leading to the disruption of the BRB and the development and progression of macular edema. Several previous studies have demonstrated that the velocity of retinal blood flow is lower in RVO patients than in subjects with normal eyes [45–47], and this reduction in blood flow velocity was correlated with the severity of RVO [48, 49]. In the present study, we also found a significant increase of V-VEGFA and a decrease in LSFG ocular blood flow at ONH (MA) in patients with RVO, and those changes were more evident in CRVO as compared to BRVO. In addition, based on the findings of this study, we conclude that FABP4 is an exclusively independent factor, and the possibility that it is synergistically involved in the pathogenesis of RVO with VEGFA cannot be excluded.

**Table 4. Stepwise multivariable regression analyses for Log V-FABP4 and Log V-VEGFA.**

| | Log V-FABP4 | | | Log V-VEGFA | |
|---|---|---|---|---|---|
| | **β** | **P** | | **β** | **P** |
| Age | 0.20 | 0.236 | Age | -0.17 | 0.282 |
| Sex (Male) | -0.24 | 0.169 | Sex (Male) | 0.41 | **0.014** |
| Log V-VEGFA | 0.35 | 0.055 | Log V-FABP4 | 0.39 | **0.021** |
| MV-MT | -0.47 | **0.015** | | | |
| | (R² = 0.437, AIC = 32) | | | (R² = 0.320, AIC = 68) | |

AIC, Akaike's information criterion; MV, mean blur rate (MBR) of the vascular area of optic nerve head (ONH); MT, MBR of the tissue area of ONH; V-FABP4, fatty acid-binding protein 4 in vitreous humor; V-VEGFA, vascular endothelial growth factor A in vitreous humor.

It has been reported that the FABP4 is considered to be primarily an adipocyte- and macrophage-specific protein, and plays an important role in maintaining glucose and lipid homeostasis [25, 27]. While the issue of why vitreous specimens obtained from patients with RVO have such high concentrations of adipocyte- and macrophage-specific FABP4 remains unclear, recent studies suggest that FABP4 is more widely expressed in a wide variety of tissues including capillaries and veins (but not arteries) and endothelial cells under normal conditions [50]. Therefore, these findings suggest that the V-FABP4 is most likely derived from retinal capillaries and venous tissue that is affected by RVO, and we therefore conclude that V-FABP4 may be pivotally involved in the regulation of the ocular blood circulation. In fact, the LSFG index of MV-MT at post-operative 1week was independently associated with Log V-FABP4 but not with Log V-VEGFA in our stepwise multivariable regression analyses. In support of this, in another study, it was reported that the velocity of retinal blood flow by LSFG was more strongly correlated with inflammatory factors than VEGF in patients with nonischemic CRVO and macular edema [51].

The pathophysiological role of FABP4 within RVO etiology has not yet been identified. However, since it was reported that VEGFA via the VEGF receptor 2 or the basic fibroblast growth factor (bFGF) induces the expression of FABP4 in endothelial cells, and in turn, FABP4 in endothelial cells promotes angiogenesis [52], our present observation that FABP4 may have significant roles within the pathogenesis of RVO with VEGF seems to be quite logical. In fact, such an effect of VEGFA on FABP4 expression could be inhibited by the chemical inhibition or siRNA knockdown of the VEGF-receptor-2. Conversely, the knockdown of FABP4 in endothelial cells significantly reduced their proliferation both under baseline conditions and in response to VEGF and bFGF. Such a suppression of the FABP4 levels can also be caused by several drugs, including a statin, eicosatetraenoic acid (EPA) / docosahexaenoic acid (DHA) agent [53], a dipeptidyl peptidase 4 inhibitor (DPP4i) [54] and an angiotensin II receptor blocker (ARB) [55]. Taken together with our present data, these observations suggest that specific inhibitors as well as neutralizing antibodies of FABP4 and antagonists of unidentified FABP4 receptors may be potential candidates for therapeutic strategies for RVO in addition to the anti-VEGF therapy.

To our knowledge, this is the first study to demonstrate the presence of V-FABP4 in patients with RVO. However, there are also several limitations that need to be considered; First, the numbers of patients enrolled in the study were relatively small (n = 32). Nevertheless, despite such small numbers in the study groups, we observed a significant positive and negative correlations between V-FABP4 and V-VEGFA (r = 0.36, P = 0.045), and V-FABP4 and MV-MT (r = -0.51, P = 0.006). Furthermore, elevation of V-VEGFA levels is a consensus observation based on a number of previous studies [56–58]. Second, the results of several statistical analyses suggest that V-FABP4 may be involved in the pathogenesis of RVO. However, the mechanisms responsible for the pathological contribution of V-FABP4 remains to be elucidated. Therefore, further investigations directed toward a better understanding of the relationship between V-FABP4, V-VEGFA and other related factors within the pathogenesis using larger numbers of patients with RVO will be needed.

## Supporting information

**S1 Table. Characteristics of the patients with non-RVO, BRVO and CRVO (n = 32).** Variables are expressed as number, means ± SD or medians (interquartile ranges). AST, aspartate transaminase; ALT, alanine transaminase; eGFR, estimated glomerular filtration rate; γGTP, γ-glutamyl transpeptidase; hsCRP, high-sensitivity C-reactive protein; MA, mean blur rate (MBR) of all area of optic verve head (ONH); MV, MBR of the vascular area of ONH; MT,

MBR of the tissue area of ONH. * P < 0.05 vs. non-RVO. † P < 0.05 vs. BRVO.
(DOCX)

## Author Contributions

**Data curation:** Megumi Watanabe, Soma Suzuki, Kaku Itoh.

**Project administration:** Masato Furuhashi.

**Writing – original draft:** Fumihito Hikage, Yosuke Ida.

**Writing – review & editing:** Hiroshi Ohguro.

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
