## [Decision Letter · Decision Letter 0]

18 Dec 2020

PONE-D-20-37537

Fatty acid-binding protein 4 is an independent factor in the pathogenesis of retinal vein occlusion

PLOS ONE

Dear Dr. Ida,

Thank you for submitting your manuscript to PLOS ONE. After careful consideration, we feel that it has merit but does not fully meet PLOS ONE’s publication criteria as it currently stands. Therefore, we invite you to submit a revised version of the manuscript that addresses the points raised during the review process.

The reviewers felt your research was performed well but had recommendations for the improvement of your paper. Please add the references mentioned by reviewer 1 and please revise your English usage and grammar as advised by reviewer 2. I recommend you find an English speaking editor to assist you.

We look forward to receiving your revised manuscript.

Kind regards,

Alfred S Lewin, Ph.D.

Academic Editor

PLOS ONE

Journal Requirements:

3.) Thank you for including your ethics statement:  "This study conformed to the principles outlined in the Declaration of Helsinki and was performed with the approval of the institutional ethical committee of our institution. Written informed consent was received from all of the participating subjects.".   

4.) Please include your tables as part of your main manuscript and remove the individual files. Please note that supplementary tables (should remain/ be uploaded) as separate "supporting information" files.

5.) Please include captions for your Supporting Information files at the end of your manuscript, and update any in-text citations to match accordingly. Please see our Supporting Information guidelines for more information: http://journals.plos.org/plosone/s/supporting-information.

6.)We noticed you have some minor occurrence of overlapping text with the following previous publication(s), which needs to be addressed:

- https://www.hindawi.com/journals/joph/2019/5185128/

In your revision ensure you cite all your sources (including your own works), and quote or rephrase any duplicated text outside the methods section. Further consideration is dependent on these concerns being addressed.

Reviewers' comments:

Reviewer's Responses to Questions

**Comments to the Author**

1. Is the manuscript technically sound, and do the data support the conclusions?

Reviewer #1: Yes

Reviewer #2: Yes

2. Has the statistical analysis been performed appropriately and rigorously? 

Reviewer #1: Yes

Reviewer #2: Yes

3. Have the authors made all data underlying the findings in their manuscript fully available?

Reviewer #1: Yes

Reviewer #2: Yes

4. Is the manuscript presented in an intelligible fashion and written in standard English?

Reviewer #1: Yes

Reviewer #2: No

5. Review Comments to the Author

Reviewer #1: I congratulate the Authors, the study was performed, with correct scientific method, and appropriate qualitative method.

The research data has been collected in a way aimed at the purpose of the research, and they sufficiently support the results.

Therefore, the research results are relevant and useful in directing future research to address the treatment and pathogenesis of retinal vein occlusiono and its complications with the suppression of the FABP4 levels, with the use of statins, drugs agents, dipeptidyl peptidase 4 inhibitors (DPP4i) and angiotensin the II receptor blockers (ARB)

The results of the present study could help give new perspectives on the subject in terms of impact on current practice.

Introduction

I recommend adding the following references to line 5 Pag. 4:

…… agents substantially improved ME and visual acuity (VA) (7

Pacella F, La Torre G, Basili S, et al. Comparison between "early" or "late" intravitreal injection of dexamethasone implant in branch (BRVO) or central (CRVO) retinal vein occlusion: six-months follow-up. Cutan Ocul Toxicol. 2017 Sep;36(3):224-230. doi: 10.1080/15569527.2016.1254648. Epub 2017 Jan 11. PMID: 27903073.

line 7 Pag. 4:

………. even though VA and ME are improved (10

Pacella E, Pacella F, La Torre G, et al. Testing the effectiveness of intravitreal ranibizumab during 12 months of follow-up in venous occlusion treatment. Clin Ter. 2012 Nov;163(6):e413-22. PMID: 23306756.

I recommend adding the following references to line 10 Pag. 4:

………. risk factor for RVO (1,

Romiti GF, Corica B, Borgi M, Visioli G, Pacella E, Cangemi R, Proietti M, Basili S, Raparelli V. Inherited and acquired thrombophilia in adults with retinal vascular occlusion: A systematic review and meta-analysis. J Thromb Haemost. 2020 Aug 17. doi: 10.1111/jth.15068. Epub ahead of print. PMID: 32805772.

Pacella F, Bongiovanni G, Malvasi M, Trovato Battagliola E, Pistone A, Scalinci SZ, Basili S, La Torre G, Pacella E. Impact of cardiovascular risk factors on incidence and severity of Retinal Vein Occlusion. Clin Ter. 2020 Nov-Dec;171(6):e534-e538. doi: 10.7417/CT.2020.2269. PMID: 33151253.

Stefanutti C, Mesce D, Pacella F, et al. Optical coherence tomography of retinal and choroidal layers in patients with familial hypercholesterolaemia treated with lipoprotein apheresis. Atheroscler Suppl. 2019 Dec;40:49-54. doi: 10.1016/j.atherosclerosissup.2019.08.031. PMID: 31818450.

Bianchi E, Scarinci F, Ripandelli G, Feher J, et al. Retinal pigment epithelium, age-related macular degeneration and neurotrophic keratouveitis. Int J Mol Med. 2013 Jan;31(1):232-42. doi: 10.3892/ijmm.2012.1164. Epub 2012 Oct 26. PMID: 23128960.

Reviewer #2: 1)Summary

The paper is appropriate in content and the research question is good. The aim of the work was to identify the fatty acid-binding protein 4 (FABP4) expressed in both adipocytes and macrophages in vitreous fluid from patients with retinal vein occlusion (RVO). The levels of FABP4 and VEGFA, ocular blood flow by laser speckle flow graphy (LSFG), height and weight, systemic blood pressures and several blood biochemistry values were collected and analyzed. The levels of V-FABP4 and V-VEGFA were higher in RVO patients compared to the non-RVO patients. The MA, MV and MV-MT were significantly decreased in CRVO patients. The correlations were observed between Log VFABP4 and Log V-VEGF, Log VFABP4 and MV-MT at post-operative 1-week, Log VFABP4 and blood urea nitrogen. The MV-MT at post-operative 1week was independently associated with Log V-FABP4 after adjustment for age and gender, and gender. The Log V-FABP4 were independently associated with Log V-VEGFA after adjustment for age.

While on the whole the paper is readable, some grammar should be checked and rewritten, and professional English language editing is required.

2)Results

Please pay attention to number, grammar and spelling mistakes, such as patients with RVO (n=20), GTP. This sentence” and ALT, GTP and hsCRP, and LSFG ocular blood folw indexes including MA” was showed as a picture format, which should be revised. In Table 1, the MT data should be checked.

3)Discussion

I would suggest rewriting the discussion section, especially second half of first paragraph which should not only repeat the results section. In addition, correlation between FABP4 and LSFG mean blur rate (MBR) values of ONH should be more discussed.

6. PLOS authors have the option to publish the peer review history of their article (what does this mean?). If published, this will include your full peer review and any attached files.

Reviewer #1: No

Reviewer #2: No

---

## [Author Response · Author response to Decision Letter 0]

31 Dec 2020

Reviewer 's Comments:

Reviewer #1: 

1) I recommend adding the following references to line 5 Pag. 4:

…… agents substantially improved ME and visual acuity (VA) (7

Pacella F, La Torre G, Basili S, et al. Comparison between "early" or "late" intravitreal injection of dexamethasone implant in branch (BRVO) or central (CRVO) retinal vein occlusion: six-months follow-up. Cutan Ocul Toxicol. 2017 Sep;36(3):224-230. doi: 10.1080/15569527.2016.1254648. Epub 2017 Jan 11. PMID: 27903073.

line 7 Pag. 4:

………. even though VA and ME are improved (10

Pacella E, Pacella F, La Torre G, et al. Testing the effectiveness of intravitreal ranibizumab during 12 months of follow-up in venous occlusion treatment. Clin Ter. 2012 Nov;163(6):e413-22. PMID: 23306756.

Answer; As pointed out, these recommended references are now included in the paper.

2) I recommend adding the following references to line 10 Pag. 4:

………. risk factor for RVO (1,

Romiti GF, Corica B, Borgi M, Visioli G, Pacella E, Cangemi R, Proietti M, Basili S, Raparelli V. Inherited and acquired thrombophilia in adults with retinal vascular occlusion: A systematic review and meta-analysis. J Thromb Haemost. 2020 Aug 17. doi: 10.1111/jth.15068. Epub ahead of print. PMID: 32805772.

Pacella F, Bongiovanni G, Malvasi M, Trovato Battagliola E, Pistone A, Scalinci SZ, Basili S, La Torre G, Pacella E. Impact of cardiovascular risk factors on incidence and severity of Retinal Vein Occlusion. Clin Ter. 2020 Nov-Dec;171(6):e534-e538. doi: 10.7417/CT.2020.2269. PMID: 33151253.

Stefanutti C, Mesce D, Pacella F, et al. Optical coherence tomography of retinal and choroidal layers in patients with familial hypercholesterolaemia treated with lipoprotein apheresis. Atheroscler Suppl. 2019 Dec;40:49-54. doi: 10.1016/j.atherosclerosissup.2019.08.031. PMID: 31818450.

Bianchi E, Scarinci F, Ripandelli G, Feher J, et al. Retinal pigment epithelium, age-related macular degeneration and neurotrophic keratouveitis. Int J Mol Med. 2013 Jan;31(1):232-42. doi: 10.3892/ijmm.2012.1164. Epub 2012 Oct 26. PMID: 23128960.

Answer; As pointed out, these recommended references are now included.

Reviewer #2: 

1) While on the whole the paper is readable, some grammar should be checked and rewritten, and professional English language editing is required.

Answer; As pointed out, the paper has been revised by a native English editing service.

2)Results

Please pay attention to number, grammar and spelling mistakes, such as patients with RVO (n=20), GTP. This sentence” and ALT, GTP and hsCRP, and LSFG ocular blood flow indexes including MA” was showed as a picture format, which should be revised. In Table 1, the MT data should be checked.

Answer; As pointed, I confirmed and changed a picture format. In addition, the MT data in Table 1 has been corrected.

3)Discussion

I would suggest rewriting the discussion section, especially second half of first paragraph which should not only repeat the results section. In addition, correlation between FABP4 and LSFG mean blur rate (MBR) values of ONH should be more discussed.

Answer; As pointed, I deleted the second half of first paragraph that included the repeating of the results section within Discussion 1st paragraph as follows; “Upon retinal hypoxia, VEGF is expressed in various cells within the retina including retinal glial cells, retinal pigment epithelial cells, and vascular endothelial cells (40), and in turn, VEGF increases vascular permeability and promotes the proliferation of endothelial cells (41, 42). In terms of the intraocular VEGF level, it was reported that those levels are not necessarily elevated in all RVO patients (43), and are widely varied among patients. In fact, it was reported that the intraocular VEGF level was significantly correlated with the retinal non-perfused area (44), as well as the severity of macular edema (43). Based upon these findings, it would appear that the expression of VEGF initially increases after RVO due to retinal hypoxia caused by a vascular occlusion, leading to the disruption of the BRB and the development and progression of macular edema. Several previous studies have demonstrated that the velocity of retinal blood flow is lower in RVO patients than in subjects with normal eyes (45-47), and this reduction in blood flow velocity was correlated with the severity of RVO (48, 49). In the present study, we also found a significant increase of V-VEGFA and a decrease in LSFG ocular blood flow at ONH (MA) in patients with RVO, and those changes were more evident in CRVO as compared to BRVO. In addition, based on the findings of this study, we conclude that FABP4 is an exclusively independent factor, and the possibility that it is synergistically involved in the pathogenesis of RVO with VEGFA cannot be excluded.”. 

In terms of correlation between FABP4 and LSFG mean blur rate (MBR) values of ONH, the corresponding discussion is included in the Discussion 2nd paragraph as follows; “It has been reported that the FABP4 is considered to be primarily an adipocyte- and macrophage-specific protein, and plays an important role in maintaining glucose and lipid homeostasis (25, 27). While the issue of why vitreous specimens obtained from patients with RVO have such high concentrations of adipocyte- and macrophage-specific FABP4 remains unclear, recent studies suggest that FABP4 is more widely expressed in a wide variety of tissues including capillaries and veins (but not arteries) and endothelial cells under normal conditions (50). Therefore, these findings suggest that the V-FABP4 is most likely derived from retinal capillaries and venous tissue that is affected by RVO, and we therefore conclude that V-FABP4 may be pivotally involved in the regulation of the ocular blood circulation. In fact, the LSFG index of MV-MT at post-operative 1week was independently associated with Log V-FABP4 but not with Log V-VEGFA in our stepwise multivariable regression analyses. In support of this, in another study, it was reported that the velocity of retinal blood flow by LSFG was more strongly correlated with inflammatory factors than VEGF in patients with nonischemic CRVO and macular edema (51).”.

---

## [Decision Letter · Decision Letter 1]

8 Jan 2021

Fatty acid-binding protein 4 is an independent factor in the pathogenesis of retinal vein occlusion

PONE-D-20-37537R1

Dear Dr. Ida,

We’re pleased to inform you that your manuscript has been judged scientifically suitable for publication and will be formally accepted for publication once it meets all outstanding technical requirements.

Kind regards,

Alfred S Lewin, Ph.D.

Section Editor

PLOS ONE

Additional Editor Comments (optional):

Reviewers' comments:

Reviewer's Responses to Questions

**Comments to the Author**

1. If the authors have adequately addressed your comments raised in a previous round of review and you feel that this manuscript is now acceptable for publication, you may indicate that here to bypass the “Comments to the Author” section, enter your conflict of interest statement in the “Confidential to Editor” section, and submit your "Accept" recommendation.

Reviewer #1: All comments have been addressed

Reviewer #2: (No Response)

2. Is the manuscript technically sound, and do the data support the conclusions?

Reviewer #1: Yes

Reviewer #2: (No Response)

3. Has the statistical analysis been performed appropriately and rigorously? 

Reviewer #1: Yes

Reviewer #2: (No Response)

4. Have the authors made all data underlying the findings in their manuscript fully available?

Reviewer #1: Yes

Reviewer #2: (No Response)

5. Is the manuscript presented in an intelligible fashion and written in standard English?

Reviewer #1: Yes

Reviewer #2: (No Response)

6. Review Comments to the Author

Reviewer #1: (No Response)

Reviewer #2: (No Response)

7. PLOS authors have the option to publish the peer review history of their article (what does this mean?). If published, this will include your full peer review and any attached files.

Reviewer #1: No

Reviewer #2: No

---

## [Editor Report · Acceptance letter]

18 Jan 2021

PONE-D-20-37537R1 

Fatty acid-binding protein 4 is an independent factor in the pathogenesis of retinal vein occlusion  

Dear Dr. Ida:

I'm pleased to inform you that your manuscript has been deemed suitable for publication in PLOS ONE. Congratulations! Your manuscript is now with our production department. 

Kind regards, 

on behalf of

Dr. Alfred S Lewin 

Section Editor

PLOS ONE